# An optimal transformation method applied to diagnose the ocean carbon budget

**Neill Mackay[1], Taimoor Sohail[2,3], Jan D. Zika** TS1 **[2,3], Richard G. Williams[4], Oliver Andrews[5], and Andrew J. Watson[1]**

[1]Faculty of Environment, Science and Economy, University of Exeter, TS2, UK

[2]School of Mathematics and Statistics, University of New South Wales, Sydney TS3, Australia

[3]Australian Centre for Excellence in Antarctic Science, University of New South Wales, Sydney TS4, Australia

[4]School of Environmental CE1 Sciences, University of Liverpool, TS5, UK

[5]Department of Environment and Geography, University of York, TS6, UK

**Correspondence:** Neill Mackay (n.mackay@exeter.ac.uk)

**Abstract.** The ocean carbon sink plays a critical role in climate, absorbing anthropogenic carbon from the atmosphere and mitigating climate change. The sink shows significant variability on decadal timescales, but estimates from models and observations disagree with one another, raising uncertainty over the magnitude of the sink, its variability, and its driving mechanisms. There is a need to reconcile observation-based estimates of air–sea $CO_2$ fluxes with those of the changing ocean carbon inventory in order to improve our understanding of the sink, and doing so requires knowledge of how carbon is transported within the interior by the ocean circulation. Here we employ a recently developed optimal transformation method (OTM) that uses water-mass theory to relate interior changes in tracer distributions to transports and mixing and boundary forcings, and we extend its application to include carbon using synthetic data. We validate the method using model outputs from a biogeochemical state estimate, and we test its ability to recover boundary carbon fluxes and interior transports consistent with changes in heat, salt, and carbon. Our results show that the OTM effectively reconciles boundary carbon fluxes with interior carbon distributions when given a range of prior fluxes. The OTM shows considerable skill in its reconstructions, reducing root-mean-squared errors from biased priors between model "truth" and reconstructed boundary carbon fluxes by up to 71 %, with the bias of the reconstructions consistently $\leq 0.06\,\mathrm{mol\,C\,m^{-2}\,yr^{-1}}$ globally. Inter-basin transports of carbon also compare well with the model truth, with residuals $< 0.25\,\mathrm{Pg\,C\,yr^{-1}}$ for reconstructions produced using a range of priors. The OTM has significant potential for application to reconcile observational estimates of air–sea $CO_2$ fluxes with the interior accumulation of anthropogenic carbon.

## 1 Introduction

The ocean is an important sink for anthropogenic carbon ($C_{anth}$), absorbing $2.9 \pm 0.4\,\mathrm{Pg\,C\,yr^{-1}}$ in the most recent decade, which represents 26 % of total emissions (Friedlingstein et al., 2023). The ocean carbon sink plays a role in mitigating atmospheric warming but at the cost of acidifying the ocean, which negatively impacts the ocean's ecosystem (Doney et al., 2009). Earth system models are sensitive to ocean carbon uptake (Arora et al., 2020), and understanding the mechanisms that govern its trends and variability is therefore crucial to the accurate projection of both future climate change and its impacts. Estimates of the ocean carbon sink from global ocean biogeochemical models (GOBMs) and surface ocean TS7 $pCO_2$-based data products show that the carbon sink has been increasing in line with increases in atmospheric $CO_2$ but with significant variability (Hauck et al., 2020; DeVries et al., 2023; Terhaar et al., 2024). The data products, which are based on the application of gap-filling methods to surface ocean $pCO_2$ observations combined with a gas transfer parameterisation, have also suggested greater decadal variability and a steeper rate of increase in the sink since the turn of the 21st century than

GOBMs, and this inconsistency poses a challenge for attempts to characterise the sink (Rödenbeck et al., 2015). According to the Global Carbon Budget (Friedlingstein et al., 2023, Table 6), the discrepancy between GOBMs and data products reached $0.6\,\mathrm{Pg\,C\,yr^{-1}}$ in 2022, or a fifth of the contemporary sink. Furthermore, while estimates of the sink from GOBMs are fairly consistent globally, differences are much larger regionally (Fay and McKinley, 2021), pointing to deficiencies in the models' representations of underlying mechanisms.

Variability in the ocean carbon sink has been linked to changes in the physical ocean circulation (DeVries et al., 2017; Caínzos et al., 2022b). Ocean circulation can impact the sink both directly (by physically transporting dissolved inorganic carbon (DIC) between the surface and deep ocean and mixing it; Bopp et al., 2015) and indirectly (by influencing changes in surface temperature, salinity, and alkalinity that control surface $p\mathrm{CO_2}$; Halloran et al., 2015). The total sink thus results from a combination of $C_{\mathrm{anth}}$ uptake driven by rising atmospheric $\mathrm{CO_2}$ concentrations and variable fluxes of both $C_{\mathrm{anth}}$ and natural carbon ($C_{\mathrm{nat}}$) driven by the redistribution of carbon and other tracers in the interior by the circulation.

A complete understanding of the ocean carbon sink cannot be attained without reconciling estimates of the air–sea flux with estimates of the changing inventory, including how carbon is redistributed in the interior. The rate of change in the global inventory of $C_{\mathrm{anth}}$ has been estimated at $2.6 \pm 0.3\,\mathrm{Pg\,C\,yr^{-1}}$ for the period 1994–2007 by Gruber et al. (2019) and $2.9 \pm 0.3\,\mathrm{Pg\,C\,yr^{-1}}$ for 1994–2004 and $2.7 \pm 0.3\,\mathrm{Pg\,C\,yr^{-1}}$ for 2004–2014 by Müller et al. (2023), with the latter estimate indicating a reduction in the ocean's carbon uptake efficiency in the more recent decade in the context of the continuing rise in atmospheric $\mathrm{CO_2}$. The inventory change estimate has also been shown to be consistent with the global air–sea flux once corrections for a preindustrial riverine outgassing and a skin temperature effect on surface $p\mathrm{CO_2}$ are taken into account (Watson et al., 2020). However, a comprehensive examination of the consistency between estimates of the air–sea flux and the interior inventory changes at the level of ocean basins has so far only been possible through the use of data-assimilating ocean biogeochemical models such as the ECCO-Darwin state estimate (Carroll et al., 2020, 2022). Such models are extremely useful in providing a mechanistic understanding of the ocean carbon sink, but they are constrained by their resolution and parameterisation of subgrid-scale processes. Another approach has used Green's functions (Haine and Hall, 2002) to describe the transport of $C_{\mathrm{anth}}$ from the surface to the interior (Khatiwala et al., 2009), but this methodology has the important caveat of assuming an ocean circulation in the steady state.

In this paper, we present a novel method of estimating air–sea $\mathrm{CO_2}$ fluxes that are consistent with changes in the ocean's carbon inventory, with interior transports, and with the mixing of carbon. The method, termed the optimal transformation method (OTM), uses a water-mass coordinate system to determine the relative roles of ocean circulation, boundary fluxes, and interior mixing in changes in the ocean's interior tracer distributions. The OTM was recently tested using outputs from a historical numerical climate model simulation and was able to recover boundary fluxes of heat and fresh water that were closer to the "true" model fluxes when biased fluxes were used as priors (Zika and Sohail, 2023). We extend the OTM's application to carbon and conduct a validation using model outputs from the ECCO-Darwin biogeochemical state estimate. OTM is an inverse method with a number of advantages over alternate approaches, enabled by its adoption of a water-mass framework that simplifies the ocean circulation with minimal loss of information. Firstly, the inverse approach can diagnose a physical tracer circulation that is consistent with observations and is not required to be in the steady state, and it can determine the transports and mixing of a tracer such as carbon by that circulation. This feature is particularly useful for the carbon sink problem, in which the non-steady-state circulation is known to play an important role in the variability (Gruber et al., 2023). Secondly, the water-mass framework allows transports and mixing consistent with boundary forcing and interior changes to be diagnosed exactly, with no need to impose a uniform vertical diffusion coefficient, as has been done with previous inverse modelling involving carbon (Caínzos et al., 2022a). Finally, the method is computationally efficient when compared to data-assimilating numerical models, while it retains sufficient spatial resolution to facilitate the analysis of mechanistic drivers of carbon sink variability. Once validated, the next phase of development will be to apply the OTM's extension to carbon to observations to produce a globally consistent estimate of ocean carbon uptake, transports, and mixing.

The remaining sections of this paper are organised as follows. In Sect. 2, an overview of the theoretical framework of the OTM is provided, including its extension to carbon. Section 3 presents the results of the validation of our method using model outputs from ECCO-Darwin, including comparisons of boundary carbon fluxes and transports of heat, fresh water, and carbon with the model "truth". Section 4 discusses the limitations of the OTM and potential challenges in its future application to observations CE2 and concludes.

## 2 Methods

### 2.1 Water-mass framework

Water-mass methods work on the principle that the properties of a water mass, for example its heat, salt, or carbon content, can fundamentally only be altered by either tracer sources and sinks or interior mixing (Groeskamp et al., 2019). For a conservative tracer, sources and sinks are limited to boundary fluxes, usually at the sea surface. Walin (1982) first proposed a framework relating temperature changes in the ocean inte-

rior to boundary heat fluxes and mixing; water-mass theory has subsequently been built upon and applied many times to studying the ocean circulation (e.g. Speer, 1993; Nurser et al., 1999; Zika et al., 2012; Groeskamp et al., 2014; Hieronymus et al., 2014; Pemberton et al., 2015; Grist et al., 2016; Evans et al., 2017; Mackay et al., 2018; Zika et al., 2021; Sohail et al., 2021).

Recently, Zika and Sohail (2023) combined aspects of a Green's function approach with water-mass theory to create the optimal transformation method. Here we briefly describe the method; for full details, the reader is referred to Zika and Sohail (2023). First, we define a set of 64 discrete water masses for the global ocean using the binary space partitioning (BSP) method of Sohail et al. (2023), splitting the upper 2000 m of the ocean into equal volumes defined between upper and lower bounds of temperature ($T$) and salinity ($S$) on an unstructured grid. The upper and lower $T$ and $S$ bounds define the water masses used for our analysis. We then further split the ocean geographically into nine basins (giving a total of 576 water masses globally) and compute the volume of each water mass in each basin, this time for the full ocean depth, as well as their mass-weighted mean $T$ and $S$ (Fig. 1). The geographical region occupied by a water mass at a given point in time is defined by a three-dimensional mask $\Omega(\mathbf{x}, t)$ TS8. We also calculate the boundary fluxes, $Q$ of heat and fresh water into each water mass, which are integrated over the outcrop area of that water mass (defined by $\Omega(x, y, 0, t)$), at each point in time (Eqs. 9 and 10). All of these quantities are calculated as monthly means using model outputs from ECCO-Darwin (see Sect. 2.3).

We then seek to relate changes in the tracer distributions from an "early" time period (with tracer concentrations $C_{0,i}$ in $N$ water masses with masses $m_{0,i}$) to a "late" time period (with tracer concentrations $C_{1,j}$ in $N$ water masses with masses $m_{1,j}$) to boundary fluxes and interior transports and mixing. Effectively, the method minimises the misfitting between an initial tracer distribution ($C_{0,i}$) and a later tracer distribution ($C_{1,j}$), taking into account the boundary fluxes $Q_{ij}^{\text{prior}}$ and the effect of the transport matrix $g_{i,j}$. We minimise a cost function which aims to find a $g_{ij}$ that is consistent with the boundary fluxes and interior changes in tracer distributions:

[Cost function] =

$$\sum_{j=1}^{N} \left\| w_j \left( \sum_{i=1}^{N} m_{0,i} g_{ij} \left( C_{0,i} + Q_{ij}^{\text{prior}} \right) - m_{1,j} C_{1,j} \right) \right\|^2. \quad (1)$$

The transport matrix represents the proportion of each early water mass $i$ that becomes part of each late water mass $j$, and $Q_{ij}^{\text{prior}}$ contains prior estimates of the boundary fluxes of each tracer that occur between the early and late time periods. When combined with the volumes and mean tracer concentrations computed in the BSP binning process (see above and Fig. 1), $g_{ij}$ allows us to determine the transports and mixing of tracers both between basins and between individual

water masses within each basin. In Eq. (1), $g_{ij}$ acts on the early tracer distribution that has been modified by the prior boundary fluxes. The weights, $w_j$, are chosen as

$$w_j = \frac{1}{A_j} \left[ \frac{1}{\text{std}(T)}, \frac{1}{\text{std}(S)} \right], \quad (2)$$

where $A_j$ represents the water-mass outcrop areas:

$$A_j = \frac{1}{t_1 - t_0} \int_{t_0}^{t_1} \int \int \Omega(x, y, 0, t) \, dx \, dy \, dt, \quad (3)$$

and $t_0$ and $t_1$ are the midpoints of the early and late time periods, respectively. The standard deviations of $T$ and $S$ are the standard deviations of the time-dependent BSP-binned water-mass mean $T$ and $S$ values. The weights effectively minimise the residual per unit outcrop area of each water mass by more strongly penalising water masses with a small outcrop in the cost function, and they normalise the contributions to the residual from different tracers. In order to avoid infinite weights where the outcrop area of a water mass is zero, the minimum value of a modified $A_j$ is set to $\min(A_j[A_j > 0])$, i.e. the smallest non-zero water-mass outcrop area. The minimisation of the cost function is subject to the following constraints:

$$0 \leq g_{ij} \leq 1; \quad (4)$$

$$m_{1,j} = \sum_{i=1}^{N} m_{0,i} g_{ij}; \quad (5)$$

$$m_{0,i} = \sum_{j=1}^{N} m_{1,j} g_{ij}; \quad (6)$$

$$g_{ij} = 0 \text{ if } \Omega_i \text{ and } \Omega_j \text{ are not in the same or adjacent basins.} \quad (7)$$

Equation (4) above ensures that the transport matrix represents a fraction of the initial water-mass volumes. Equations (5) and (6) impose the conservation of mass for the sum of all the water masses. Equation (7) limits the geographical range of water-mass interactions, excluding the possibility of unrealistic tracer transport. We solve the optimisation problem for $g_{ij}$ with the Python *cvxpy* package, using the "MOSEK" solver with default settings.

## 2.2 Extension to carbon

In the above, we outlined our method in the context of its application to two conservative tracers: temperature and salinity. In order to extend its application to studying the ocean carbon sink, we use the tracer $C^*$, first proposed by Gruber et al. (1996) and defined as

$$C^* = \text{DIC} - R_{\text{C:P}} \text{PO}_4 - 0.5(\text{ALK} + R_{\text{N:P}} \text{PO}_4), \quad (8)$$

where DIC, $\text{PO}_4$, and ALK are the dissolved inorganic carbon concentration, phosphate concentration, and alkalinity,

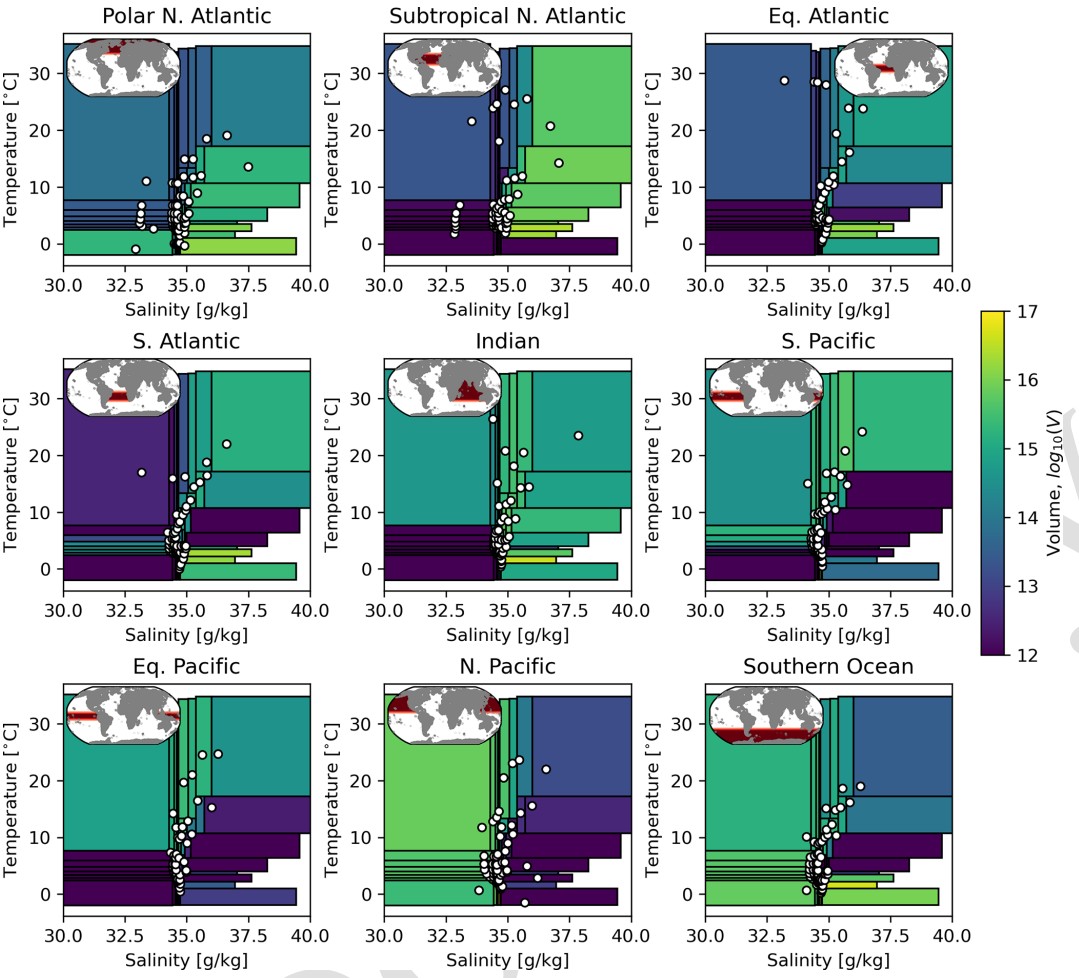

**Figure 1.** Water-mass definitions from the binary space partitioning (BSP) of ECCO-Darwin (early-period time average for 1995–2005). For each of nine ocean basins, the edges of the boxes show the global definitions of each water mass in temperature–salinity space, and the colours show the volume of water occupied by that water mass in that basin. The white dots show the volume-weighted mean temperature and salinity of each water mass for each basin. The inset at the top left of each subplot shows the geographical basin definition.

respectively, and $R_{C:P}$ and $R_{N:P}$ are the C : P and N : P stoichiometric ratios, respectively. $C^*$ is quasi-conservative in the ocean. In a model with fixed stoichiometric ratios such as ECCO-Darwin, it should be exactly conservative. As such, within the context of this study, $C^*$ has the same property as $T$ and $S$: within a water mass, it can only be modified by either boundary fluxes or mixing. We use the fixed ratios from ECCO-Darwin of $R_{C:P} = 120$ and $R_{N:P} = 16$. Figure 2 shows the changes in $C^*$ in $T$–$S$ space between an early period, taken as the time mean of the ECCO-Darwin tracer distributions from 1995–2005, and a late period, taken as the time mean of the tracer distributions from 2005–2015. The differences in the distributions of $T$, $S$, and $C^*$ between these two periods and the boundary fluxes of those three tracers form the basis for the application of the OTM in this study. Note that it is the *transition* between the state of the ocean in the early period and its state in the later one that we use to infer the carbon uptake and transport. In this case, the

OTM solution could be regarded as representing an average for the time period between the midpoint of 1995–2005 and the midpoint of 2005–2015 (i.e. for the change from 2000 to 2010). In setting up our inverse problem (Eq. 1), we use the same water masses, defined in $T$–$S$ coordinates, as plotted on Fig. 1, with the mass-weighted mean $C^*$ in each water mass incorporated as additional elements of $\mathbf{C}$.

We test two distinct implementations of the incorporation of carbon into the OTM. In the first implementation, there is no prior estimate of the boundary carbon flux, and the optimisation to minimise the cost function in Eq. (1) is carried out by inputting (i) the concentrations of $T$ and $S$ for the two time periods and (ii) the boundary fluxes $Q_{ij}^{\text{prior}}$ for heat and fresh water, which are calculated from the ECCO-Darwin surface

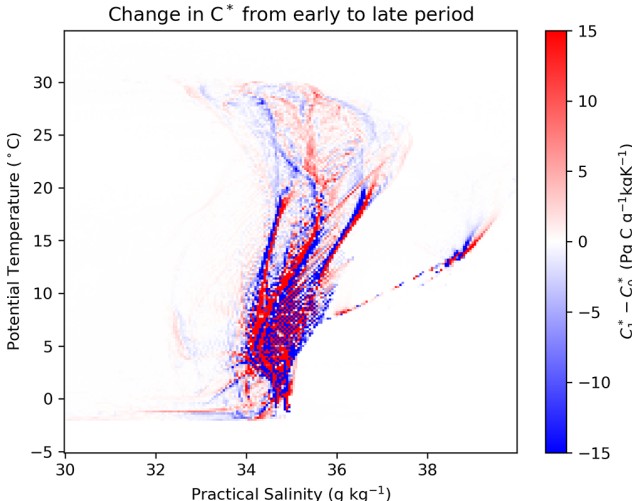

**Figure 2.** Changes in the amount of the conservative tracer $C^*$ (see Sect. 2.2) per unit temperature and salinity in ECCO-Darwin from the early averaging period (1995–2005) to the late averaging period (2005–2015). Red indicates an increase in carbon in that region of $T$–$S$ space; blue indicates a decrease.

forcing as follows:

$$Q_{i,T}^{\text{prior}} = \frac{1}{\rho_0 C_p m_{0,i}(t_1 - t_0)} \times \int_{t_0}^{t_1} \int \int \Omega_i(x, y, 0, t) q_T(x, y, t) \, dx \, dy \, dt \tag{9}$$

$$Q_{i,S}^{\text{prior}} = \frac{1}{\rho_0 m_{0,i}(t_1 - t_0)} \int_{t_0}^{t_1} \int \int \Omega_i(x, y, 0, t) \times (q_S(x, y, t) - S_0 q_{\text{FW}}(x, y, t) + q_{\text{SP}}) \, dx \, dy \, dt. \tag{10}$$

Here, $\rho_0$ is a reference density, $1029 \, \text{kg m}^{-3}$; $C_p$ is the heat capacity of seawater, $4000 \, \text{J kg}^{-1} \text{K}^{-1}$; $q_T$ is the surface heat flux in $\text{W m}^{-2}$; $q_S$ is the surface salt flux in $\text{g m}^{-2} \text{s}^{-1}$; $S_0$ is a reference salinity, $35 \, \text{g kg}^{-1}$; $q_{\text{FW}}$ is the surface freshwater flux in $\text{kg m}^{-2} \text{s}^{-1}$; and $q_{\text{SP}}$ is the depth-summed salt tendency due to the salt plume flux from sea ice formation in $\text{g m}^{-2} \text{s}^{-1}$. In this case, the transport ~~solution~~ $g_{ij}$ represents transports that are consistent with the changing $T$ and $S$ distributions and the surface heat and salt- or freshwater boundary fluxes. We impose constraints on the transport matrix $g_{ij}$ for the volume transport through the Bering Strait, which is set to 1.1 Sv northwards ($1 \, \text{Sv} = 10^6 \, \text{m}^3 \text{s}^{-1}$), and the Indonesian throughflow, which is set to a net transport of 15 Sv westwards, based on volume transports from ECCO-Darwin. Having obtained the transport matrix $g_{ij}$, we then estimate the boundary carbon flux from the residual between the final $C^*$ distribution, $C_{1,j}$, and the initial $C^*$ distribution, $C_{0,i}$, modified by the transport $g_{ij}$:

$$Q_j^{\text{adjust}} = C_{1,j} - \frac{1}{m_{1,j}} \sum_{i=1}^{N} m_{0,i} g_{ij} C_{0,i}. \tag{11}$$

Note that for obtaining the $C^*$ adjustment, we use the BSP-binned mean carbon concentrations for $C$ in Eq. (11); we can equally obtain the adjustments in $T$ and $S$, but these mismatches are small since we have used the exact boundary forcings for heat and fresh water as priors.

In the second implementation, we include a prior estimate of the boundary carbon flux (which we modify with estimates of the uncertainty; see Sect. 2.3) and include all three tracers ($T$, $S$, and $C^*$) in the minimisation of Eq. (1). In this case, the weights are

$$w_j = \frac{1}{A_j} \left[ \frac{1}{\text{std}(T)}, \frac{1}{\text{std}(S)}, \frac{1}{\text{std}(C^*)} \right], \tag{12}$$

and the unmodified carbon flux is

$$Q_{i,C^*}^{\text{prior}} = \frac{1}{m_{0,i}(t_1 - t_0)} \times \int_{t_0}^{t_1} \int \int \Omega_i(x, y, 0, t) q_{\text{CO}_2}(x, y, t) - \Omega_i(x, y, D, t) q_{\text{sed}}(x, y, t) \, dx \, dy \, dt, \tag{13}$$

where $q_{\text{CO}_2}$ is the air–sea $CO_2$ flux, $D$ is the water depth at $(x, y)$, and $q_{\text{sed}}$ is the sediment flux of $C^*$ due to falling particulate matter, which, in ECCO-Darwin, is removed from the model when particulate matter hits the sea floor. The mask $\Omega_i$ is applied such that $q_{\text{CO}_2}$ acts at the surface and $q_{\text{sed}}$ acts at the sea floor. The sediment flux is calculated as

$$q_{\text{sed}} = \text{POC} w_{\text{POC}} - R_{\text{C:P}} \text{POP} w_{\text{POP}} - \frac{R_{\text{N:P}}}{2} \text{POP} w_{\text{POP}}, \tag{14}$$

where POC and POP are the particulate organic carbon and particulate organic phosphorus concentrations, respectively, in the bottom wet grid cell of the model; $w_{\text{POC}} = w_{\text{POP}} = 10 \, \text{m d}^{-1}$ TS9 are the sinking rates for particulate organic carbon and particulate organic phosphorus, respectively; and $R_{\text{C:P}}$ and $R_{\text{N:P}}$ are stoichiometric ratios as in Eq. (8). For this implementation, the OTM seeks a transport matrix $g_{ij}$ that is consistent with the prior fluxes and interior changes in $T$, $S$, and $C^*$, minimising the adjustment for all three tracers as diagnosed from Eq. (11).

## 2.3 Validation with numerical-model output

The ECCO-Darwin model (Carroll et al., 2020) is an ocean biogeochemical model based on the ECCO state estimate (Forget et al., 2015) that is coupled online to the MIT Darwin ecosystem model (Dutkiewicz et al., 2015). ECCO uses an adjoint of the CE3 MITgcm to assimilate all available physical observations into an internally consistent tracer-conserving

estimate of the physical ocean state that matches well with the observational data while obeying the model's dynamical equations. Meanwhile, the Darwin component is optimised using a Green's function approach to obtain the best fit to biogeochemical observations. ECCO-Darwin is run from 1992–2017, and we use monthly mean outputs from January 1995–December 2015 for validating our optimal transformation method. The model has 50 vertical levels and uses the ECCO LLC270 horizontal grid, which has 13 tiles of $270 \times 270$ grid cells each, and a horizontal grid spacing of between $1/3°$ at the Equator and $\sim 18$ km at high latitudes (Carroll et al., 2020).

An important feature of ECCO-Darwin that makes it a good choice for this study is that the budgets of heat, fresh water, and $C^*$ are closed over the time period covered by the model outputs we are using. Confirming that these budgets close globally provides a good check that the principle underpinning the OTM – that only boundary fluxes or interior mixing can change water-mass properties for conservative tracers – will hold. The heat, fresh water, and carbon budgets computed from the BSP-binned values that we input into the OTM are shown in Fig. A1 TS10. There are small offsets in the heat and freshwater budgets because we have used monthly mean fields, which introduces a temporal error. In the carbon budget, there is a budget residual of around 3 % that accumulates over the second half of the time period. This residual is most likely due to slight differences between our calculation of the sediment flux of $C^*$, which is not a model variable but which we calculate from Eq. (8), and the loss of $C^*$ to the model's bottom boundary. In any case, the residual is smaller than we might hope to achieve when applying our method to observations in a future study. This residual places a lower limit on the residual that we can expect from our OTM optimisations.

In order to assess the ability of OTM to obtain a consistent estimate of the uptake, transports, and mixing of carbon from imperfect information, we test five cases with different priors for the boundary carbon flux. The first case uses no prior, as already outlined in Sect. 2.2. Case 2 uses the ECCO-Darwin model fluxes binned into water-mass space (Eq. 13), which is as close to the model "truth" as can be achieved with this method (see Fig. 5). For cases 3–5, we construct a 2D field of uncertainties in the air–sea $CO_2$ flux based on an ensemble of observational estimates compiled by Fay et al. (2021), which combines six different observationally based $pCO_2$ data products with five different wind products via a gas transfer parameterisation to produce 30 different air–sea $CO_2$ flux estimates. We calculate an observational uncertainty as the standard deviation of the time mean of these 30 estimates at every grid point, and we use that as the basis for uncertainties in our $CO_2$ flux priors. The gridded uncertainties are shown in Fig. A2. For cases 3 and 4, we add a negative and a positive bias with a magnitude of $2\times$ the observational uncertainties (i.e. $2\sigma$ TS11 or the 95 % confidence interval) to the ECCO-Darwin air–sea $CO_2$ flux

at each grid point before computing the BSP-binned $Q_{i,C^*}^{\text{prior}}$ using Eq. (13), giving us lower and upper bounds (Fig. 4g and j). For case 5, a bias of $2\times$ the observational uncertainty but with the same sign as the ECCO-Darwin flux is applied, meaning that fluxes are biased in the direction of the prior flux (either into or out of the ocean; Fig. 4m).

## 2.4 Remapping into geographical coordinates

The information contained in the priors and raw OTM solutions is organised according to the 576 water masses defined in the BSP binning process. To aid interpretation, we map the solutions for the carbon fluxes and their priors back into geographical coordinates using a time average over the early period (1995–2005) of the mask used for the BSP binning, such that $q(x, y, z)$ is the carbon flux in Cartesian coordinates:

$$q(x, y, z)^{\text{prior}} = \sum_{i=1}^{N} \frac{1}{(t_1 - t_0)} Q_i^{\text{prior}} \Omega_i(x, y, z) \qquad (15)$$

$$q(x, y, z)^{\text{mix}} = \sum_{j=1}^{N} \frac{1}{(t_1 - t_0)} Q_j^{\text{mix}} \Omega_i(x, y, z) \qquad (16)$$

$$q(x, y, z)^{\text{adjust}} = \sum_{j=1}^{N} \frac{1}{(t_1 - t_0)} Q_j^{\text{adjust}} \Omega_i(x, y, z), \qquad (17)$$

where

$$Q_j^{\text{mix}} = \frac{1}{m_{1,j}} \left[ \sum_{i=1}^{N} m_{0,i} g_{ij} \left( C_{0,i} + Q_i^{\text{prior}} \right) \right] \\ - C_{0,i=j} - Q_{i=j}^{\text{prior}} \qquad (18)$$

is the effect of transports and mixing on each tracer. The depth-integrated carbon flux at each grid point is calculated as

$$q(x, y) = \sum_{z} \left[ q^{\text{prior}}(x, y, z) + q^{\text{adjust}}(x, y, z) \right], \qquad (19)$$

i.e. the sum over all depths of the prior plus the adjustment for the carbon fluxes. Note that Eq. (19) calculates the depth integral of the carbon flux remapped from water-mass space using three-dimensional masks (Eqs. 15 and 17) and results from a combination of the air–sea flux and the sediment flux. This depth-integrated carbon flux is not the same as the air–sea flux acting on the surface water-mass outcrops; the three-dimensional remapping is performed to ensure that our remapped fields include all sources and sinks of $C^*$ since not all water masses with an associated sediment flux have a surface outcrop. Our aim is to assess the ability of the OTM to reconstruct the ECCO-Darwin boundary fluxes for a closed carbon budget.

## 3 Results

### 3.1 Carbon sink

The skill of the OTM is assessed for five different cases with different priors and cost functions as described in Sect. 2.3. For case 1, where the OTM is given no prior carbon flux and the cost function is minimised by only considering the changes in the $T$ and $S$ distributions and their associated boundary fluxes, the agreement between the OTM solution and the ECCO-Darwin model truth is generally good when integrated over each basin (Fig. 3a). The exceptions are in the polar North Atlantic, where the OTM overestimates the uptake, and in the North Pacific, where it is underestimated. When remapped into geographical coordinates, it is more obvious where the OTM has struggled to converge towards the model truth with limited information (Fig. 4a and b). The RMSE between the model truth and the prior or solution reduces from 0.89 to 0.76 $\mathrm{mol\,C\,m^{-2}\,yr^{-1}}$, and the bias reduces from $-0.36$ to 0.07 $\mathrm{mol\,C\,m^{-2}\,yr^{-1}}$.

For case 2, where the OTM is given an ideal prior carbon flux based on the model truth and the cost function is minimised for $T$, $S$, and $C^*$, the basin-integrated solution is close to the model truth for all basins (Fig. 3b). For the remapped fluxes (Fig. 4d and e), there is a small region in the Sea of Japan where the OTM has drifted slightly away from the model truth, but the solution matches the model very closely across the rest of the ocean. The RMSE between the model truth and the prior or solution increases slightly from 0.00 to 0.04 $\mathrm{mol\,C\,m^{-2}\,yr^{-1}}$, and the bias increases fractionally from 0.00 to $-0.01$ $\mathrm{mol\,C\,m^{-2}\,yr^{-1}}$.

For cases 3 and 4, where the prior carbon fluxes represent the 95th percentile lower and upper bounds, respectively, based on an observational uncertainty estimate, the adjustment away from the priors towards the model truth in the basin-integrated OTM solutions is striking (Fig. 3c and d). Examining the remapped fluxes for case 3, the bias is reduced across most of the ocean, with small negative biases remaining in the Southern Ocean and western subpolar North Atlantic and positive biases remaining in the subtropical North Atlantic and the Sea of Japan (Fig. 4h). The RMSE reduces by almost two-thirds from 0.59 to 0.22 $\mathrm{mol\,C\,m^{-2}\,yr^{-1}}$, and the bias nearly disappears, going from $-0.41$ to $-0.02$ $\mathrm{mol\,C\,m^{-2}\,yr^{-1}}$. Case 4 shows the greatest improvement from prior to solution, with the RMSE reducing from 0.59 to 0.18 $\mathrm{mol\,C\,m^{-2}\,yr^{-1}}$ and the bias from 0.41 to 0.02 $\mathrm{mol\,C\,m^{-2}\,yr^{-1}}$. Similar to case 3, there are patches of bias in the solution in the North Atlantic and the Sea of Japan.

Case 5 tests the limits of the method's capability: basin-integrated fluxes are close to the model truth and tend to be moving away from the prior and towards the truth in the basins with larger fluxes and, especially, globally (Fig. 3e). However, in the equatorial Pacific, the basin-integrated solution is further from the truth than the prior, and the remapped fluxes do not show an obvious improvement from prior to solution (Fig. 4m and n). The RMSE reduces slightly from 0.33 to 0.29 $\mathrm{mol\,C\,m^{-2}\,yr^{-1}}$, and the bias reduces from 0.12 to 0.05 $\mathrm{mol\,C\,m^{-2}\,yr^{-1}}$.

The remapped OTM solutions from cases 2–5 are very similar to each other across most of the ocean (Fig. 4f, i, l, and o) and to the remapped model truth, shown on Fig. 5b. The latter gives a visual representation of the best theoretical solution that can be obtained with this method when binning each of nine basins into 64 water masses using the BSP binning process and then remapping back into geographical coordinates. The original model fluxes are shown in Fig. 5a for comparison. The case 1 solution (Fig. 4c) is broadly similar to the model truth in terms of the main regions of uptake and outgassing, indicating that $T$ and $S$ constraints alone provide a good first guess for the redistribution of carbon as a means of estimating the boundary carbon fluxes when combined with the interior changes in $C^*$. This is then improved by the addition of the prior boundary carbon flux $Q_{i,C^*}$ and the inclusion of carbon in the cost function (Eq. 1) in cases 2–5.

### 3.2 Meridional tracer transports

We next assess our OTM solutions with respect to meridional transports of heat, fresh water, and carbon at the boundaries between our nine ocean basins. These are obtained for the OTM solutions by combining the transport matrix $g_{ij}$ with the time means over the early period of the water-mass mean tracer values calculated during the BSP binning process, i.e. $T_0$, $S_0$, and $C_0^*$, which are also modified by their prior boundary fluxes:

$$[\text{Heat transport}]^{\mathrm{OTM}} =$$
$$C_p \rho_0 \sum_{i=1}^{N} m_{0,i} \left( T_{0,i} + Q_{i,T}^{\mathrm{prior}} \right) g_{ij}\delta_{ij} \tag{20}$$
$$[\text{Freshwater transport}]^{\mathrm{OTM}} =$$
$$\frac{-\rho_0}{S_0} \sum_{i=1}^{N} m_{0,i} \left( S_{0,i} + Q_{i,S}^{\mathrm{prior}} \right) g_{ij}\delta_{ij} \tag{21}$$
$$[\text{Carbon transport}]^{\mathrm{OTM}} =$$
$$\sum_{i=1}^{N} m_{0,i} \left( C_{0,i}^* + Q_{i,C^*}^{\mathrm{prior}} \right) g_{ij}\delta_{i,j}, \tag{22}$$

where $\delta_{ij} = 1$ if the water mass $i$ is upstream of the boundary between two basins and $j$ is downstream, $\delta_{ij} = -1$ if $j$ is upstream and $i$ is downstream, and $\delta_{ij} = 0$ for unconnected basins. ECCO-Darwin provides comparable meridional transports from the residual between the latitude-

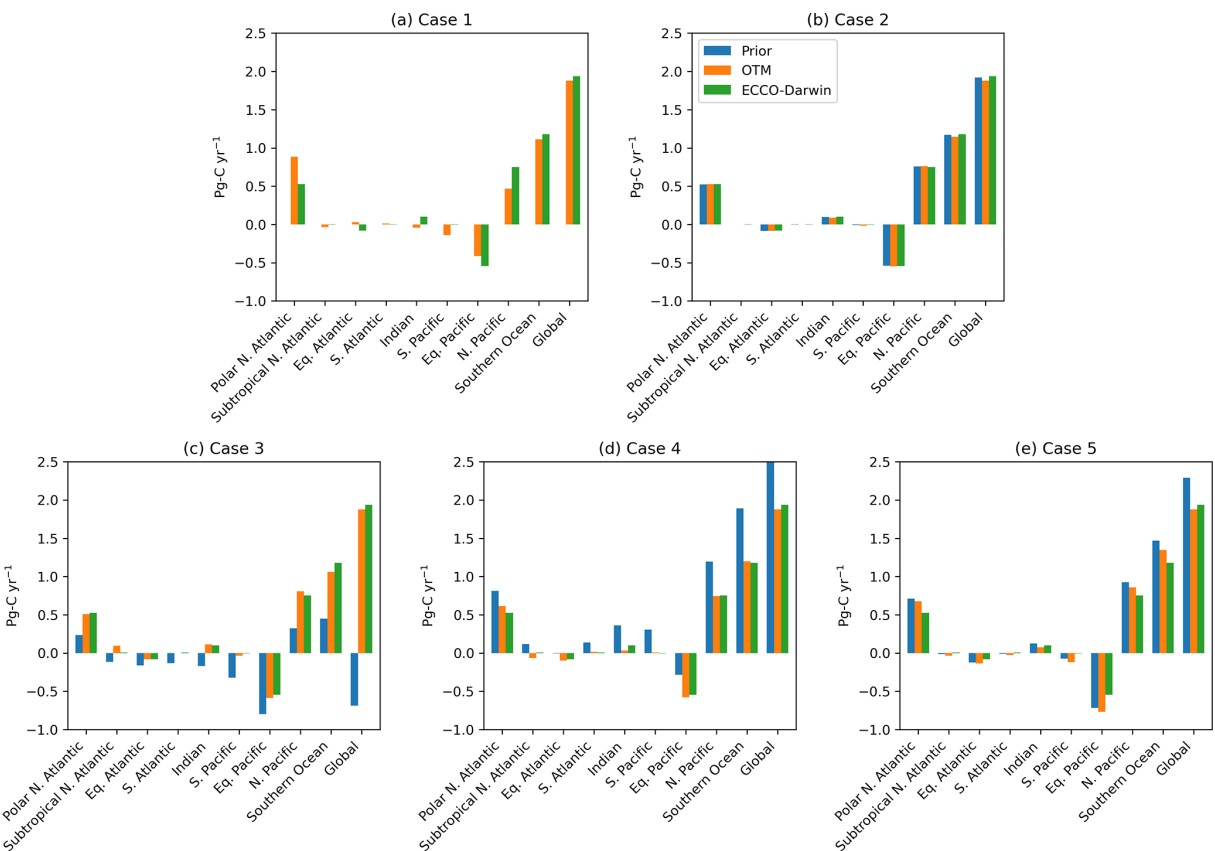

**Figure 3.** Basin-integrated carbon fluxes (positive into the ocean) from five cases with different priors. Blue bars show the prior fluxes, orange bars show the OTM solutions, and green bars show the ECCO-Darwin model truth. Note that in panel **(d)**, the prior global flux is $4.5\,\mathrm{Pg\,C\,yr^{-1}}$; the $y$ axis is cut off for ease of comparison with the other cases.

integrated air–sea fluxes and interior changes:

$[\text{Heat transport}]^{\text{E-D}}(y') =$

$$\frac{1}{t_1 - t_0} \int\limits_{t_0}^{t_1} \int\limits_{y=0}^{y'} \int q_T(x,y,t)\,\mathrm{d}x\mathrm{d}y\mathrm{d}t$$

$$-\frac{C_p \rho_0}{t_1 - t_0} \int \int\limits_{y=0}^{y'} \int [T(x,y,z,t_1) - T(x,y,z,t_0)]\,\mathrm{d}x\mathrm{d}y\mathrm{d}z \qquad (23)$$

$[\text{Freshwater transport}]^{\text{E-D}}(y') = \dfrac{-\rho_0}{S_0(t_1 - t_0)}$

$$\times \int\limits_{t_0}^{t_1} \int\limits_{y=0}^{y'} \int \left[ q_S(x,y,t) - \frac{S_0}{\rho_0} q_{\text{FW}}(x,y,t) + q_{\text{SP}} \right] \mathrm{d}x\mathrm{d}y\mathrm{d}t$$

$$+ \frac{\rho_0}{S_0(t_1 - t_0)}$$

$$\times \int \int\limits_{y=0}^{y'} \int [S(x,y,z,t_1) - S(x,y,z,t_0)]\,\mathrm{d}x\mathrm{d}y\mathrm{d}z \qquad (24)$$

$[\text{Carbon transport}]^{\text{E-D}}(y') =$

$$\frac{1}{t_1 - t_0} \int\limits_{t_0}^{t_1} \int\limits_{y=0}^{y'} \int \left[ q_{\text{CO}_2}(x,y,t) - q_{\text{sed}}(x,y,t) \right] \mathrm{d}x\mathrm{d}y\mathrm{d}t$$

$$-\frac{1}{t_1 - t_0}$$

$$\times \int \int\limits_{y=0}^{y'} \int \left[ C^*(x,y,z,t_1) - C^*(x,y,z,t_0) \right] \mathrm{d}x\mathrm{d}y\mathrm{d}z. \qquad (25)$$

The volume integrations in Eqs. (23)–(25) are applied globally and then separately for the Atlantic and Indo-Pacific using a mask. For the Atlantic and Indo-Pacific comparisons, the OTM values calculated from Eqs. (20)–(22) have the northward transport of heat, fresh water, or carbon that flows through the Bering Strait subtracted from them. This correction is necessary because the Atlantic mask includes the Arctic Ocean, whereas the Indo-Pacific mask excludes it, which means that the integrations implicitly assume that all of the residual between the boundary fluxes and interior changes results in meridional transports that flow through the Atlantic, when in reality the transports are split between the two.

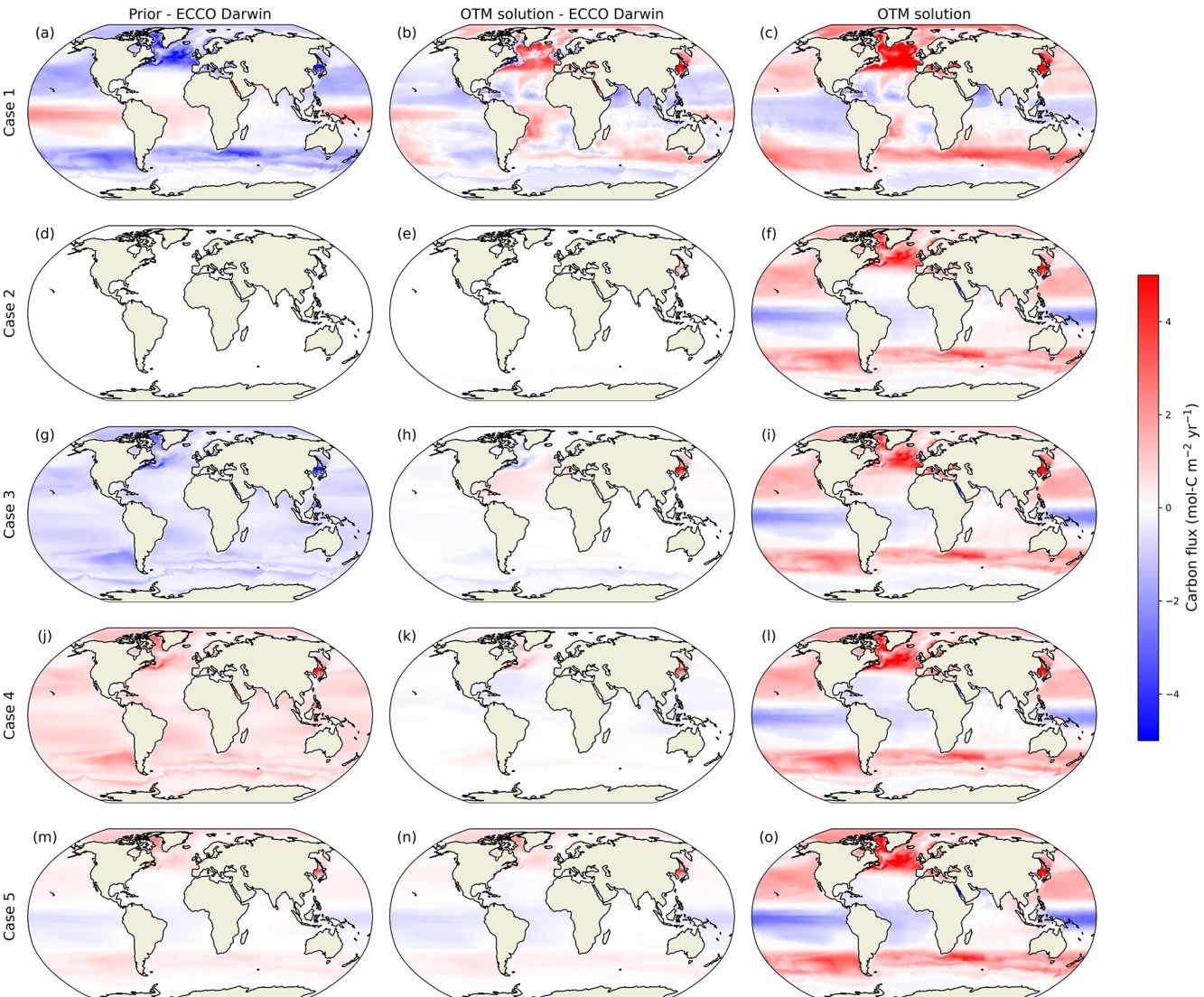

**Figure 4.** OTM solutions from five cases with different carbon flux priors, remapped into geographical space. The left column **(a, d, g, j, m)** shows the carbon flux difference between the prior and the ECCO-Darwin model truth, the middle column **(b, e, h, k, n)** shows the difference between the OTM solution and the model truth, and the right column **(c, f, i, l, o)** shows the OTM solution. The top row **(a–c)** shows the solution for case 1 with no carbon flux prior; the second row **(d–f)** shows case 2, where a prior based on the model truth was used; rows 3 **(g–i)** and 4 **(j–l)** show cases 3 and 4, where the prior fluxes included a bias of 2 times an a-priori uncertainty calculated at each grid point; and the bottom row **(m–o)** shows case 5, where double the a-priori uncertainty was applied as a bias with the same sign as the prior flux (see Sect. 2.3). Red and blue indicate either positive and negative biases compared to the model truth, respectively **(a, b, d, e, g, h, j, k, m, n)**, or fluxes into and out of the ocean, respectively **(c, f, i, l, o)**.

A comparison between the meridional transports of heat, fresh water, and carbon obtained from the OTM solution for case 2 and ECCO-Darwin is shown in Fig. 6. For this experiment, the OTM transports are very close to the model truth for all three tracers at almost all latitudes where we can make comparisons at inter-basin boundaries, both globally and separately for the Atlantic and Indo-Pacific. There is a residual of $\sim 0.2$ Sv for freshwater transport at 10° N in the Indo-Pacific. We can assume that there is a much smaller residual at the latitude at which transport zero-crossing oc-

curs since the green line passes over the position of the green dot just to the south of it. The heat and freshwater transports from the OTM solutions for cases 1, 3, 4, and 5, which are not plotted, are nearly identical to those for case 2 in Fig. 6; however, there is some variation in the solutions for carbon (Fig. 7). The discrepancies between OTM and ECCO-Darwin are fairly small for cases 3 and 4, but for case 1, although the transports match well globally at 10° N and 35° S, the southward transport in the Atlantic is significantly overestimated and the Indo-Pacific transports are offset north-

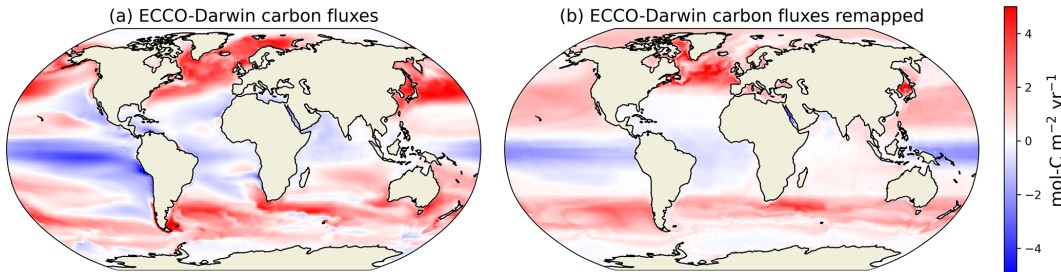

**Figure 5.** Boundary carbon fluxes (air–sea CO$_2$ flux CE4 – sediment flux) for the ECCO-Darwin 1995–2015 time mean **(a)** and the BSP-binned fluxes remapped back into geographical coordinates **(b)**. Note that **(b)** is the ECCO Darwin flux against which the prior and solution are compared in Fig. 4; it is also the prior for case 2.

wards by a similar amount. This mismatch indicates that the OTM is unable to recover the correct transports of carbon solely from information about the changes in temperature and salinity and associated boundary fluxes of heat and salt water or fresh water, and it shows that the additional information provided by the a priori CO$_2$ flux estimates in cases 2–5 is needed. Nonetheless, the reasonable agreement between the basin-integrated OTM carbon fluxes and the model truth for case 1 in Fig. 3a is encouraging since it suggests that, even with limited information, the OTM gets the divergences in carbon transports approximately correct. For case 5, slightly too much southward transport in the Northern Hemisphere in both the Atlantic and Indo-Pacific causes the global mismatch at 10° N to be worst for this experiment. At 35° S, the Indo-Pacific and global northward transports are similarly overestimated by the OTM. Overall, the spread in the estimates of meridional tracer transports across cases 2–5 is $< 0.30$ Pg C yr$^{-1}$, and the mismatch between the OTM solutions and the model truth is $< 0.25$ Pg C yr$^{-1}$.

## 3.3  Basin carbon budgets

The net inter-basin carbon transports from the OTM solution for case 2 are shown in Fig. 8, along with the boundary carbon fluxes (air–sea flux – sediment flux). The overall picture is of a convergence of meridional carbon transports and accompanying surface outgassing between 10° N and 10° S and a divergence of transports and accompanying surface uptake at higher latitudes. A large counter-clockwise circulation of carbon between the Southern Ocean, the South Pacific, and the Indian Ocean is likely an instance where the departure of the OTM solution from reality is not evident from the zonally integrated comparisons in Fig. 6; we will discuss this further in Sect. 4.1. Integrating the transport divergences, boundary fluxes, and interior changes allows us to construct a carbon budget for each basin (Fig. 9 and Table A1). The largest changes occur in the polar North Atlantic, equatorial Pacific, North Pacific, and Southern Ocean. In the polar North Atlantic, an uptake of 0.52 Pg C yr$^{-1}$ is balanced by a roughly even split between a net transport of $-0.29$ Pg C yr$^{-1}$ out of the basin and an increase in the basin's inventory

of 0.24 Pg C yr$^{-1}$. In the equatorial Pacific, a transport of 0.70 Pg C yr$^{-1}$ into the basin is balanced by an outgassing of $-0.55$ Pg C yr$^{-1}$ and a small increase in the basin's inventory of 0.15 Pg C yr$^{-1}$. In the North Pacific, roughly the reverse happens, with an uptake of 0.76 Pg C yr$^{-1}$ balanced by a transport out of the basin of $-0.51$ Pg C yr$^{-1}$ and an inventory increase of 0.25 Pg C yr$^{-1}$. Finally, in the Southern Ocean, an uptake of 1.14 Pg C yr$^{-1}$ goes mainly into an inventory increase of 0.79 Pg C yr$^{-1}$, with a smaller transport of $-0.36$ Pg C yr$^{-1}$ occurring out of the basin. Globally, the budget is closed, with an uptake of 1.88 Pg C yr$^{-1}$ balanced by an equal increase in inventory.

## 3.4  Interior remapping of OTM solutions

So far, we have examined the OTM solutions in terms of a depth-integrated view of exchanges between basins and boundary forcings. We next analyse the three-dimensional remapped fields from Eqs. (15)–(17) to explore the contribution of boundary fluxes versus that of transports and mixing to changes in the ocean interior. Zonal mean sections of these components and their sum, which (by construction) equals the inventory change, for the Atlantic and Pacific are shown for the top 1500 m in Fig. 10 and for the full ocean depth in Fig. A3. Note that remapping the components using the 3D mask $\Omega_i(x, y, z)$ means that the carbon changes are averaged over the full geographical extent of each water mass in the ocean interior, which implies the assumption that each water mass is homogeneous, i.e. that they have been well mixed over the 10-year timescale under consideration. At high northern latitudes in the Atlantic, boundary fluxes cause an invasion of carbon into water masses that penetrate to greater depths moving equatorwards, with the maximum occurring around 40° N (Fig. 10a). A similar pattern appears in the South Atlantic, with additional penetration to $\sim 500$ m depth occurring near Antarctica. Meanwhile, outgassing water masses in the lower latitudes are confined to the top $\sim 200$ m. In the Pacific, patterns are similar, but there is greater penetration to depth in the Southern Hemisphere, and the outgassing layer reaches deeper but over a narrower range of latitudes (Fig. 10b). Transports and mixing move

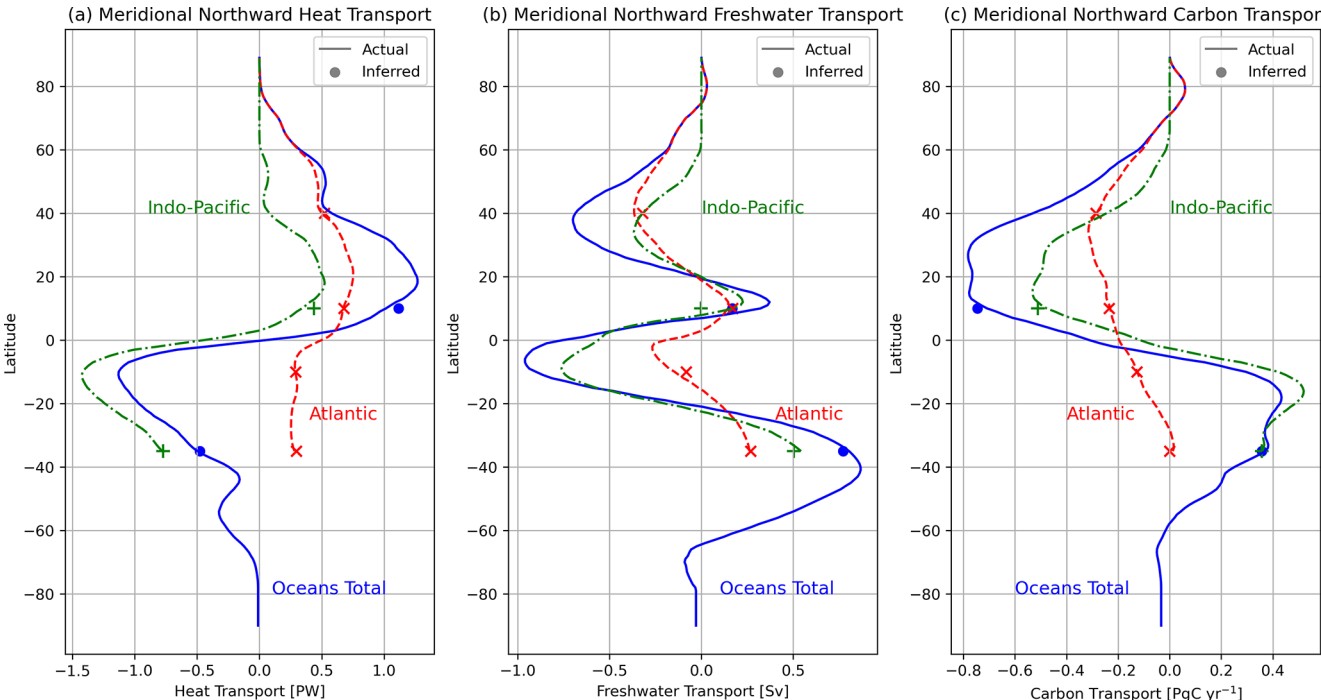

**Figure 6.** Zonally integrated meridional transports of **(a)** heat, **(b)** fresh water, and **(c)** carbon from ECCO-Darwin at all latitudes (lines) and their equivalents from the OTM solution for case 2 at the latitudes of the inter-basin boundaries (point markers). The globally integrated transports are shown in blue (solid lines and dots), the transports integrated over the Indian and Pacific oceans are shown in green (dashed–dotted lines and crosses), and the transports integrated over the Atlantic Ocean are shown in red (dashed lines and multiplication symbols).

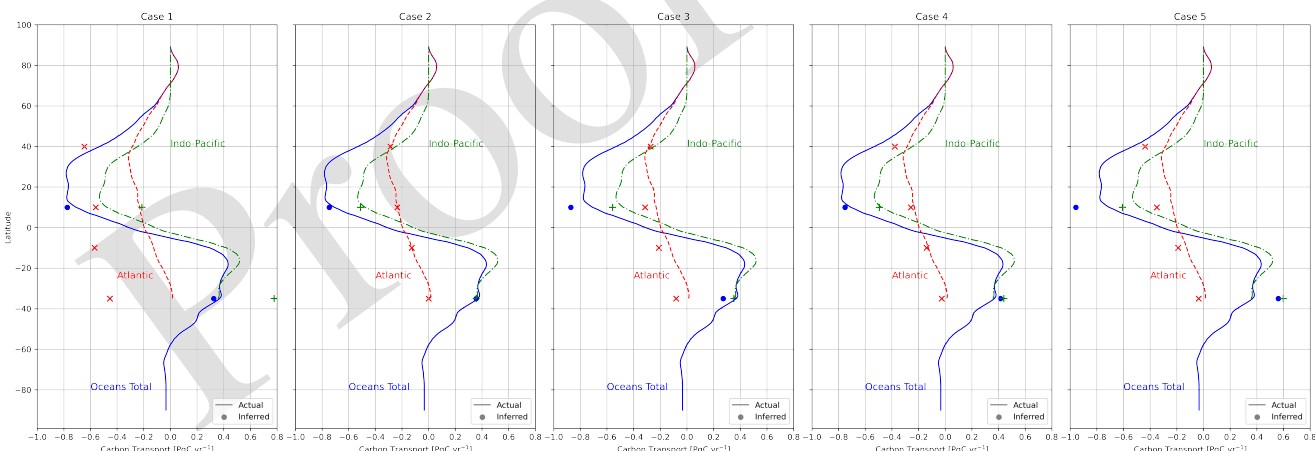

**Figure 7.** Zonally integrated meridional transports of carbon from ECCO-Darwin at all latitudes (lines) and their equivalents from OTM solutions for cases 1–5 at the latitudes of the inter-basin boundaries (point markers). The globally integrated transports are shown in blue (solid lines and dots), the transports integrated over the Indian and Pacific oceans are shown in green (dash–dotted lines and crosses), and the transports integrated over the Atlantic Ocean are shown in red (dashed lines and multiplication symbols).

carbon away from high latitudes, where the atmosphere provides a carbon source, and towards shallower waters at low latitudes in both the Atlantic and Pacific, where the atmosphere provides a carbon sink; hence, transports and mixing oppose the effect of the surface flux (Fig. 10c and d). In the Atlantic, low-latitude transports and mixing move car-

bon away from the region just below the surface, dominating the interior change and causing a net loss of carbon between $\sim 40°$ N and $\sim 30°$ S and at $\sim 150$–$700$ m depth (Fig. 10c and e). In the Pacific, transport dominates in the subtropical cells, causing a net loss of carbon between $\sim 30°$ S and $\sim 40°$ N and CE5 up to $\sim 500$ m depth; this signal may be due

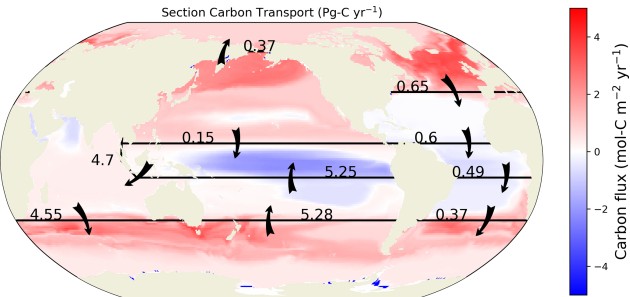

**Figure 8.** Inter-basin net transports of carbon (black arrows) and the boundary carbon flux (background colours; positive into the ocean) for the OTM solution for case 2. Transports are in $\mathrm{Pg\,C\,yr^{-1}}$.

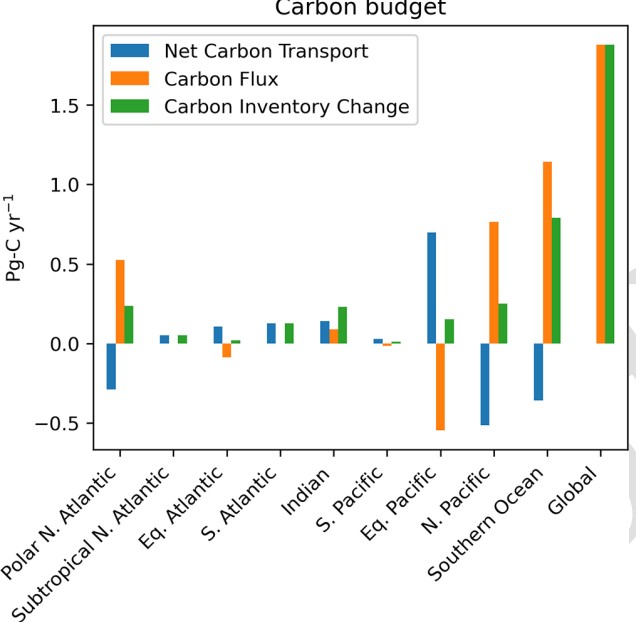

**Figure 9.** Carbon budget in each basin for the OTM solution for case 2. Blue bars show the net carbon transports, orange bars show the carbon flux (positive into the ocean), and green bars show the inventory change.

to wind-driven subduction of carbon from the mixed layer. In the Atlantic, transports and mixing between $\sim 750$–$2000\,\mathrm{m}$ depth cause net accumulation in the equatorial and northern subpolar regions (Fig. A3c and e). In the deeper waters, transports and mixing lead to a reduction in carbon concentrations between $\sim 40°\,\mathrm{N}$ and $\sim 30°\,\mathrm{S}$ below $\sim 3000\,\mathrm{m}$ depth (Fig. A3c and e); this signal is likely an imprint of the Atlantic Meridional Overturning Circulation, which upwells in the Southern Ocean before returning northwards and supplies carbon from the deep ocean that is later outgassed at lower latitudes.

# 4 Discussion

## 4.1 Limitations of the optimal transformation method

The optimal transformation method may be used to diagnose the interior transports and mixing of carbon consistent with boundary fluxes and interior changes and to recover improved (i.e. closer to the truth) boundary fluxes from biased priors when this method is applied to model data and perfect information is available. When the given information is limited to changes in temperature and salinity distributions and their boundary forcings, the OTM obtains a transport matrix that is broadly consistent with changes in carbon and can be used to obtain a reasonable basin-integrated carbon uptake. However, inter-basin meridional carbon transports from the OTM are inconsistent with the model truth when using this setup, indicating that more information is needed for a realistic solution. With the addition of prior information about the distribution of boundary carbon fluxes, the OTM shows considerable skill in recovering carbon fluxes that are closer to the model truth than the prior while also diagnosing inter-basin carbon transports that are consistent with the model. We now discuss some limitations of this method and the caveats and considerations that are relevant to its future application to observations.

A fundamental limitation of the OTM stems from its use of a water-mass coordinate system, from which it also derives its utility. The use of this system necessitates some loss of information compared with, for example, the state estimation product we have used for its validation. Through the BSP binning process, we average tracer concentrations and their boundary fluxes over water masses, and the method therefore does not resolve either tracer or flux gradients within a water mass. The effect of the latter is illustrated in Fig. 5, where we compare the unaltered ECCO-Darwin boundary carbon fluxes with the result of binning the fluxes in water-mass space and then remapping them back into geographical coordinates using a mask (Eqs. 15, 17, and 19). There are differences; for example, off the west coast of North America, ECCO-Darwin has outgassing (Fig. 5a) and the remapped fluxes show uptake (Fig. 5b); a similar situation occurs off the west coast of the southern tip of South America and in the northern Indian Ocean. Figure 5b is the closest we can get to the true model fluxes with this configuration (with the caveat, noted in Sect. 2.4, that we have remapped surface fluxes into three-dimensional water masses). Note that a closer match to the "true" field may be achieved by combining the remapped term $q(x, y, z)^{\mathrm{adjust}}$ with the true fluxes; therefore, here we present a worst-case scenario where the detail for the true fluxes is not assumed to be well known. In this validation, we divided each of nine basins into 64 water masses chosen as a compromise between spatial resolution and required computing resources. In water-mass coordinates, OTM is extremely computationally cheap by comparison with data-assimilation or Green's function approaches, with solutions to the opti-

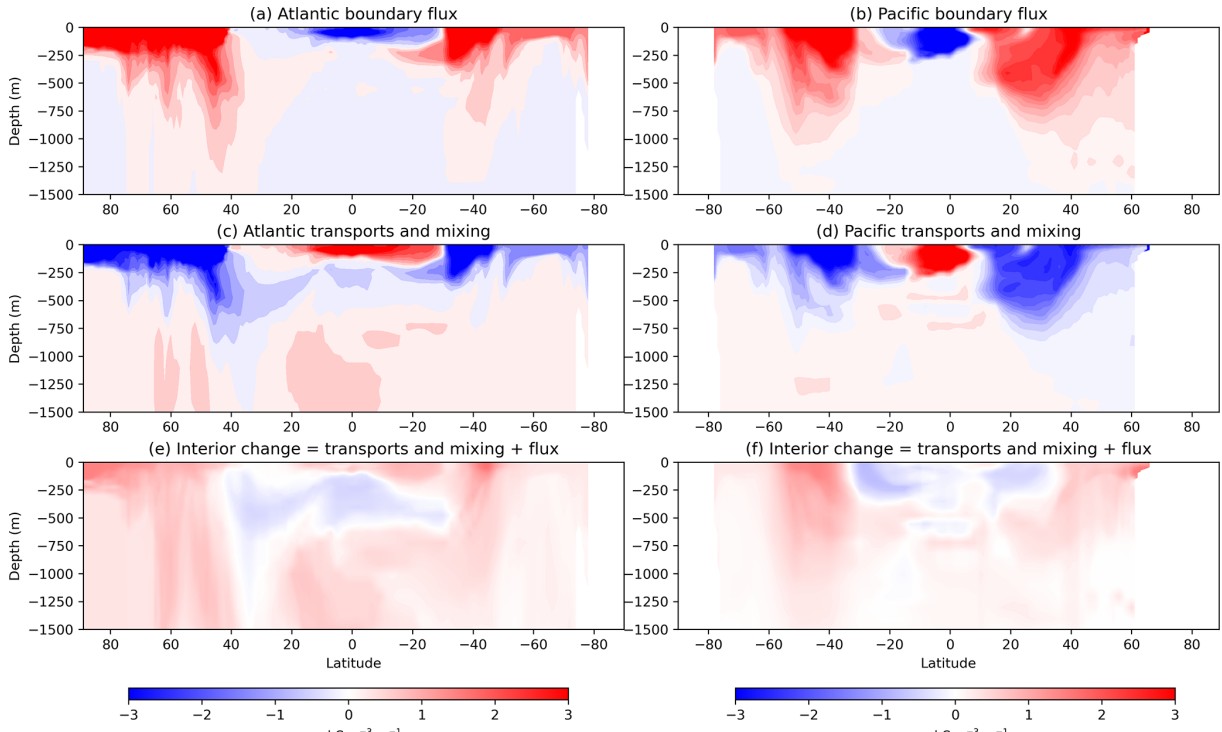

**Figure 10.** Zonally averaged interior carbon concentration changes from the OTM solution for case 2 in the Atlantic (left column: **a, c, e**) and the Pacific (right column: **b, d, f**) for the top 1500 m of the ocean. The top row (**a, b**) shows the impact of carbon fluxes (air–sea flux + sediment flux) in or out of the ocean on the carbon concentrations; the middle row (**c, d**) shows the impact of transports and mixing; and the bottom row (**e, f**) shows the interior changes, which are the sum of the first two rows. Positive values represent increases in carbon concentration between our early (1995–2005) and late (2005–2015) averaging periods due to each component, and negative values are decreases over that time.

misation problem obtained within a few seconds. However, the remapping back into geographical coordinates to aid the interpretation of the results is more intensive, requiring the storage of large files containing the mask $\Omega(\mathbf{x}, t)$, and this places limits on the number of water masses that it is feasible to define.

In general, beyond the mass conservation constraints in Eqs. (5) and (6), mass transports are unconstrained in the OTM; instead, it is the conservation of tracers that provides the main constraint on the solutions. This tracer constraint means that solutions may be obtained that are consistent with changes in temperature, salinity, and carbon but that are physically unrealistic in other ways. For example, if the OTM needs to cool down a particular region through a flux of cold water, the method can achieve that by flowing either a small amount of very cold water or a larger amount of moderately cold water into that region. Or, similarly, the carbon concentration can be increased somewhere by some amount; this increase might be achieved through either a small flux of extremely carbon-rich water or a larger flux of less carbon-rich water. A possible impact of the lack of mass transport constraints is seen in the large counter-clockwise circulation of carbon between the Southern Ocean, South Pacific,

and Indian Ocean in Fig. 8. These transports do not seem consistent with, for example, a westward transport $C_{\mathrm{anth}}$ of 0.05 PgC yr$^{-1}$ due to the Indonesian Throughflow (ITF), as reported by Mikaloff Fletcher et al. (2006), or an eastward transport $C_{\mathrm{nat}}$ of 0.1 PgC yr$^{-1}$ due to the ITF, as reported by Mikaloff Fletcher et al. (2007). We have diagnosed the transport $C^*$, which contains both $C_{\mathrm{anth}}$ and a portion of $C_{\mathrm{nat}}$, but the OTM transports do not appear to be reconcilable with the ITF estimates. In a future application, it may prove necessary to impose additional constraints on inter-basin mass transports; for example, in the case of the South Pacific or the Indian Ocean, it could be beneficial to further split the Southern Ocean in a manner that allows the imposition of an Antarctic Circumpolar Current. Another avenue would be to impose some quasi-vertical structure on the inter-basin mass transports, which could be done in temperature coordinates, salinity coordinates, or both.

We have placed subjective constraints on the connectivity between our nine basins such that all water masses in neighbouring basins are able to mix with one another and water masses from basins that are not neighbours cannot mix. This constraint might be too permissive in some places (for example, it allows water masses from the equatorial North Pacific

to mix with those in the subtropical North Atlantic) or too strict in others (for example, it forbids water masses from the equatorial Atlantic that might be carried northwards in the western boundary current from mixing with those in the polar North Atlantic). The fidelity of such constraints is also dependent on the timescale under consideration, which in our case was the decade between the early and late averaging periods but could be shorter or longer.

Two further assumptions that have been subjectively imposed concern the boundary forcings. The first is that we have used exact heat and freshwater forcings from ECCO-Darwin, thereby implicitly assuming that these are known. Zika and Sohail (2023) explored the OTM's ability to recover boundary forcings of heat and fresh water from biased priors, and our focus was on the uptake, the transport and mixing, and the storage of carbon. However, when applying this technique to observations, there will be uncertainties in both the boundary forcings and the interior changes for all three tracers, and these errors will need to be considered. The boundary fluxes for heat and fresh water could be explored using multiple products such as the ERA5 reanalysis (global coverage from 1940 to present; Hersbach et al., 2020) and the JRA55 reanalysis (global coverage from 1958 to present; Kobayashi et al., 2015), and those for $CO_2$ could be explored with the compilation of data products from Fay and McKinley (2021) that were used to assess the flux uncertainty in Sect. 2.3. The second assumption relates to the order of action of boundary fluxes versus interior transports and mixing. In a forward model such as ECCO-Darwin, time steps are discrete but comparatively short, such that boundary fluxes and interior transports and mixing occur effectively simultaneously, as they do in the real ocean. In the OTM framework, we have to make a choice about whether to apply the boundary forcing first and then calculate the transports and mixing required for the tracers to reach their late distributions, whether to perform these steps in the opposite order, or whether to use some combination of these two approaches. In this study, we experimented with the first two possibilities for carbon, with the setups for our cases 2–5 adopting the former (fluxes then mixing) and case 1 adopting the latter (mixing then fluxes), and we found that the former produced the best results.

## 4.2 Future application

One more component is necessary for the future application of the OTM to observations for the purpose of studying ocean carbon that we are yet to discuss: the inventory change. The temperature and salinity in the ocean are comparatively well observed, and their time evolutions in the ocean interior have been mapped in a gridded product with monthly global fields spanning 1900 to present, based on a combination of shipboard and Argo float observations in the Met Office EN4 objective analysis (Good et al., 2013). By contrast, the ocean's interior carbon concentrations are considerably more sparsely sampled, with re-occupations of oceanographic sections collecting carbonate system variables usually taking place around once per decade (Gruber et al., 2023). Recently, products based on the first efforts to reconstruct the time history of ocean interior carbon have emerged. These include a neural-network technique similar to that employed by Landschützer et al. (2013) to map surface $pCO_2$, which in this case was applied to DIC from the GLODAP database (Olsen et al., 2019) to produce the MOBO-DIC climatology (Keppler et al., 2020) and its time-varying successor (Keppler et al., 2023). Another approach, which also used machine learning, by Zemskova et al. (2022) extrapolated from satellite data by combining it with numerical model output. Unfortunately, these two estimates are limited, respectively, to the top 1500 m of the ocean and the Southern Ocean. MOBO-DIC could form the basis for an application of the OTM, but it would be necessary to somehow extend it to full depth. A third technique developed by Turner et al. (2023) has been demonstrated as being capable of reconstructing ocean interior carbon by ensemble optimal interpolation using only the relationships between carbon, temperature, salinity, and atmospheric $CO_2$ from models, but no product applying this method to observations has become available yet. The method does, nonetheless, demonstrate the plausibility of accurately reconstructing interior carbon from relationships to the better-observed temperature and salinity fields, consistent with the findings from our case 1 experiment. We are developing our own global, full-depth, time-evolving reconstructions of DIC and $C^*$ in the ocean from 1990 to present, which we hope to combine with the OTM in future work. The reconstructions use deep neural networks trained on GLODAP DIC, total alkalinity, and nutrient data, with predictors of temperature and salinity from EN4, location, depth, and atmospheric $CO_2$ concentration.

## 5 Conclusion

We have presented the application of a novel optimal transformation method (OTM) to diagnose the uptake, transport, and storage of carbon in the global ocean. The method utilises a balance between boundary forcings and interior transports and mixing in water-mass space for conservative tracers, and we have validated it using outputs from the ECCO-Darwin biogeochemical state estimate. When given prior estimates of the boundary forcing for carbon with biases based on reasonable observational uncertainties, the OTM was able to recover the true carbon forcing and also to diagnose interior transports and the mixing of heat, fresh water, and carbon that were consistent with the model truth. When applied to observational reconstructions of changes in ocean carbon, the OTM has the potential to reconcile changes in the interior anthropogenic carbon inventory with air–sea $CO_2$ fluxes through the action of physical transports and mixing.

## Appendix A

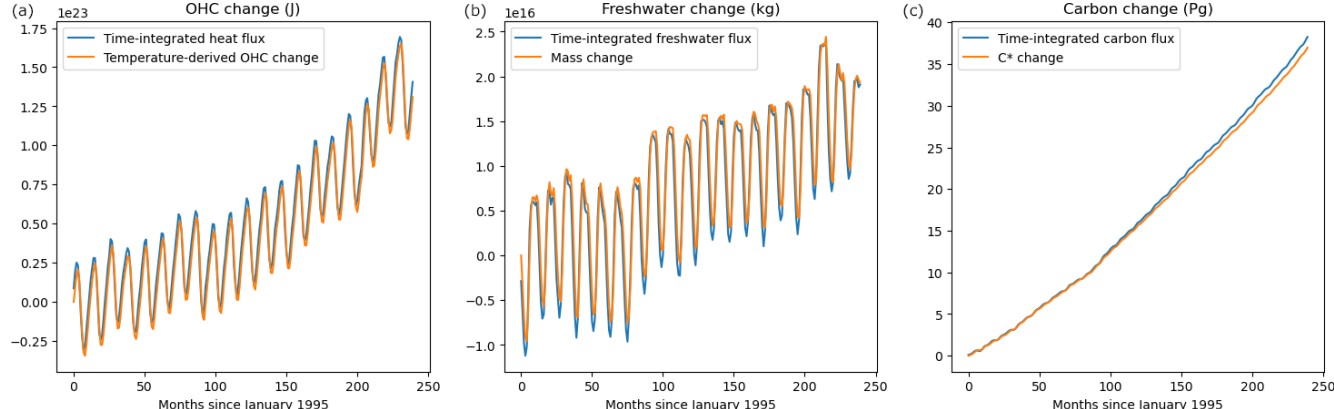

**Figure A1.** Budgets of ocean heat content (OHC; **a**), fresh water (**b**), and carbon (**c**) resulting from the BSP binning of ECCO-Darwin outputs for input to the OTM. The sum of the boundary forcings is shown in blue, and the interior changes inferred from changes in $T$, $S$, and $C^*$ are shown in red.

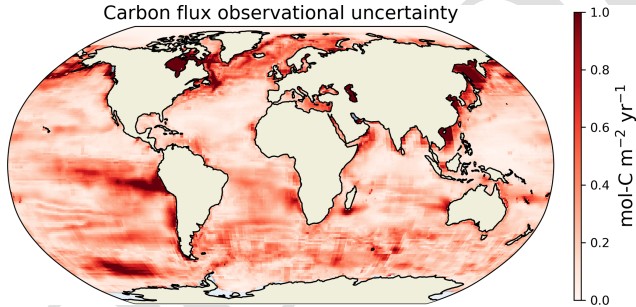

**Figure A2.** Observational uncertainties added to the carbon flux priors as described in Sect. 2.3.

**Table A1.** Carbon budget for the OTM solution for case 2, as plotted in Fig. 9. Values are in $\mathrm{Pg\,C\,yr^{-1}}$.

|                          | Transport | Flux  | Inventory change |
| ------------------------ | --------: | ----: | ---------------: |
| Polar N. Atlantic        |    −0.29  |  0.52 |             0.24 |
| Subtropical N. Atlantic  |     0.05  |  0.00 |             0.05 |
| Eq. Atlantic             |     0.11  | −0.09 |             0.02 |
| S. Atlantic              |     0.13  |  0.00 |             0.13 |
| Indian                   |     0.14  |  0.09 |             0.23 |
| S. Pacific               |     0.03  | −0.02 |             0.01 |
| Eq. Pacific              |     0.70  | −0.55 |             0.15 |
| N. Pacific               |    −0.51  |  0.76 |             0.25 |
| Southern Ocean           |    −0.36  |  1.14 |             0.79 |
| Total                    |     0.00  |  1.88 |             1.88 |

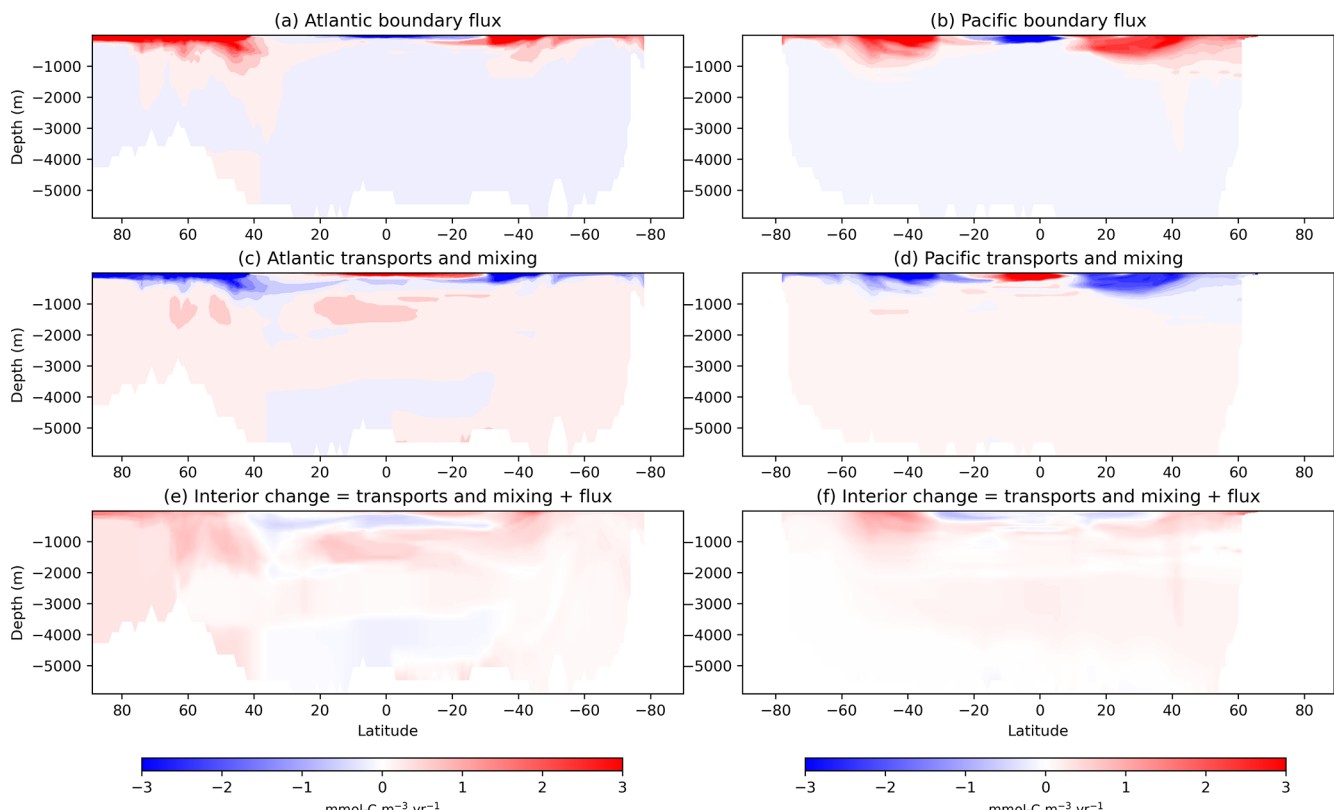

**Figure A3.** Full-depth zonally averaged interior carbon concentration changes from the OTM solution for case 2 in the Atlantic (left column: **a, c, e**) and the Pacific (right column: **b, d, f**). The top row (**a, b**) shows the impact of carbon fluxes (either air–sea or sediment fluxes) in or out of the ocean; the middle row (**c, d**) shows the impact of transports and mixing on the carbon concentrations; and the bottom row (**e, f**) shows the interior changes, which are the sum of the first two rows. Positive values represent increases in carbon concentrations between our early (1995–2005) and late (2005–2015) averaging periods due to each component, and negative values represent decreases over that time.

*Code and data availability.* The ECCO-Darwin model data used for the validation of the optimal transformation method are available for download from https://data.nas.nasa.gov/ecco/index.html (last access: TS12). The code used for producing and plotting the results presented here is available on Zenodo via the DOI https://doi.org/10.5281/zenodo.10782587 (Mackay, 2024) TS13.

*Author contributions.* The optimal transformation method was conceived by JDZ and developed by JDZ and TS. NM implemented its extension to carbon and carried out the validation performed in this study, with input from JDZ, TS, AJW, RGW, and OA. NM prepared the manuscript, adapting code created by TS to produce some of the figures. All authors contributed to the final text.

*Competing interests.* The contact author has declared that none of the authors has any competing interests.

ther geographical representation in this paper. While Copernicus Publications makes every effort to include appropriate place names, the final responsibility lies with the authors.

*Acknowledgements.* This work was supported by the National Environmental Research Council grant NE/W001543/1. Andrew Watson is additionally supported by the Royal Society research professorship grants RP/EA/180008 and RP140106. For the purpose of open access, the author has applied a 'TS14 Creative Commons attribution (CC BY) licence to any author-accepted manuscript version arising.

*Financial support.* This research has been supported by the NAME OF FUNDER (grant no. GRANT AGREEMENT NO). TS15

*Review statement.* This paper was edited by Vassilios Vervatis and reviewed by two anonymous referees.

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

**Remarks from the language copy-editor**

CE1  Please note that the spelling of "Sciencies" in affiliation 4 was corrected.

CE2  "and concludes" means "and ends". Did you actually mean "and draws conclusions" here? Please check.

CE3  Please define "MITgcm" for clarity.

CE4  Should this dash be a minus sign, or do you mean the air–sea $CO_2$ flux and sediment flux combined here? Please clarify the meaning of the dash here (and in the first sentence of the "Basin carbon budgets" section).

CE5  Does "up to" mean "above" or "below" here? Please clarify.

**Remarks from the typesetter**

TS1  Please confirm middle names in Jan D. Zika, Richard G.Williams, and Andrew J.Watson.

TS2  Please provide city and zip code.

TS3  Please provide zip code.

TS4  Please provide zip code.

TS5  Please provide city and zip code.

TS6  Please provide city and zip code.

TS7  Please confirm "p" is a variable.

TS8  Please check throughout the text that all vectors are denoted by bold italics and matrices by bold roman.

TS9  Unit has been changed to exponential format.

TS10  The composition of Fig. A1 has been adjusted to our standards.

TS11  Please confirm "$2\sigma$".

TS12  Please check URL and provide DOI if possible. Please also provide a corresponding reference list entry.

TS13  Please confirm citation.

TS14  Unpaired single quote. Please check.

TS15  Please note that there is funding information given in the acknowledgements, but you did not indicate any funding upon manuscript registration. Therefore, we were not able to complete the financial support statement. Please provide the missing information and double-check your acknowledgements to see whether repeated information can be removed from the acknowledgements. Thanks.

TS16  Please ensure that any data sets and software codes used in this work are properly cited in the text and included in this reference list. Thereby, please keep our reference style in mind, including creators, titles, publisher/repository, persistent identifier, and publication year. Regarding the publisher/repository, please add "[data set]" or "[code]" to the entry (e.g. Zenodo [code]).

TS17  Please provide page range or article ID.

TS18  Please provide page range or article ID.

TS19  Please provide page range or article ID.

TS20  Please confirm reference list entry.

TS21  Please provide page range or article ID.

TS22  Please provide more details regarding this publication depending on the reference category.

TS23  Please provide the publisher.

TS24  Please provide page range or article ID.

TS25  Please provide volume number.

TS26  Please provide page range or article ID.

TS27  Please update if possible.