# Peer review of "An optimal transformation method applied to diagnosing the ocean carbon budget"

_EGUsphere, 2023_

## Author Response (AR1)

**An optimal transformation method applied to diagnosing the ocean carbon sink – response to review**

We thank the reviewer for their considerate review of our manuscript. Please see our responses below in blue (original comments in black).

The results, here based on the ECCO model, are convincing and I wondered how this could be applied to real world with observations. Quoting authors: "Once validated, OTM's extension to carbon can be applied to observations to produce a globally consistent estimate of ocean carbon uptake, transports and mixing. ». It would be useful to inform the observations needed and that could be used for applying the OTM (data, periods, region…) and derived carbon change at global scale.

We have added the following text to the last paragraph of section 4.1: "The boundary fluxes for heat and freshwater could be explored using multiple products such as the ERA5 reanalysis (global coverage from 1940 to present; Hersbach et al. (2020)) and the JRA55 reanalysis (global coverage from 1958 to present; Kobayashi et al. (2015)), and for $CO_2$ with the compilation of data products by Fay and McKinley (2021) that were used to assess the flux uncertainty in section 2.3."

We also added a detail in the second sentence of section 4.2, which now reads: "Temperature and salinity in the ocean is comparatively well observed, and their time evolution in the ocean interior has been mapped based on a combination of shipboard and Argo float observations in the Met Office EN4 objective analysis (Good et al., 2013), **in a gridded product with monthly global fields spanning 1900-present.**"

Finally, we added some more detail about the carbon reconstructions in section 4.2, as outlined below in response to C-09.

C-01: Line 23: "…but with significant variability (Hauck et al., 2020)." Maybe also refer to De Vries et al (2023) and Terhaar et al (2024).
We have added the suggested references.

C-02: Line 25: "…have also suggested greater decadal variability and a steeper rate of increasing sink since the turn of the 21st century than GOBMs,". Maybe recall that the difference could reach 1 PgC/yr (compared to 2.9 PgC/yr listed line 7).
We added the following sentence, including a reference to the latest GCB paper: "According to the Global Carbon Budget (Friedlingstein et al., 2023, Table 6), the discrepancy between GOBMs and data products reached 0.6 Pg C yr$^{-1}$ in 2022, or a fifth of the contemporary sink." We also updated the reference in line 7 and added the uncertainty to the contemporary sink estimate.

C-03: Line 36: "The rate of change of the global inventory of Canth has been estimated at 2.6 ± 0.3 PgCyr−1 for the period 1994-2007 (Gruber et al., 2019)". Maybe also refer to Müller et al (2023) who estimate change from 1994 to 2004 by 29 ± 3 PgC/decade but to 27 ± 3 PgC/decade from 2004 to 2014 (i.e. a weakening of the uptake ?). Also, I think these estimates where calculated for the layer 0-3000m only, not the full depth, and these results should be extended to the bottom (as proposed using OTM).

Thank you for the suggestion. The Gruber and Müller estimates are, in fact, full depth, with a scaling factor applied to derive the accumulation below 3000 m. We have altered the text to "The rate of change of the global inventory of $C_{anth}$ has been estimated at 2.6 ± 0.3 Pg C yr$^{-1}$ for the period 1994-2007 by Gruber et al. (2019), and 2.9 ± 0.3 Pg C yr$^{-1}$ for 1994-2004 and 2.7 ± 0.3 Pg C yr$^{-1}$ for 2004-2014 by Müller et al. (2023), with the latter estimate indicating a reduction in ocean's carbon uptake efficiency in the more recent decade in the context of the continuing rise in atmospheric $CO_2$."

C-04: Line 290: « This mismatch indicates that OTM is unable to recover the correct transports of carbon solely from information about the changes in temperature and salinity and associated boundary fluxes of heat and salt/freshwater ». This is an important result suggesting that for carbon one need to use apriori fluxes as well (correct ?).

Yes, that's correct! We have modified the text to "This mismatch indicates that OTM is unable to recover the correct transports of carbon solely from information about the changes in temperature and salinity and associated boundary fluxes of heat and salt/freshwater, and that the additional information provided by the a priori $CO_2$ flux estimates in cases 2-5 is needed."

We have also emphasised this point at the start of the discussion: "When given information limited to changes in temperature and salinity distributions and their boundary forcings, OTM obtains a transport matrix that is broadly consistent with changes in carbon, and which can be used to obtain reasonable basin-integrated carbon uptake. However, inter-basin meridional carbon transports from OTM are inconsistent with the model truth using this setup, indicating that more information is needed for a realistic solution. With the addition of prior information about the distribution of boundary carbon fluxes, OTM shows considerable skill in recovering carbon fluxes that are closer to the model truth than the prior, while also diagnosing inter-basin carbon transports consistent with the model."

C-05: Line 298: « The net inter-basin carbon transports from the case 2 OTM solution are shown on Fig. 8… ». Figure 8 and 9 show the total carbon transport for 1995-2015; could you also show the same for the difference between 1995-2005 and 2010-2015 to highlight the power of OTM to derive carbon budget changes?

In fact we are not able to plot trends with the current setup, because our solution derives from the *transition* between the time average of the early period (1995-2005) and the time average of the late period (2005-2015). We have added some text to make this clearer: "Note that it is the *transition* between the state of the ocean in the early period and its state in the later one that we use to infer the carbon uptake and transport. In this case, the OTM solution could be regarded as representing an average for the time period between the midpoint of 1995-2005 and the midpoint of 2005-2015 (i.e. for the change from 2000 to 2010)."

C-06: Figure 8: Compared to the carbon transport from Mikaloff Fletcher et al (2007), there is a difference of the carbon flux in the Indian Ocean and at the Indonesian throughflow. On Line 145 you informed that "the Indonesian throughflow is set to a net transport of 15 Sv westwards, based on volume transports from ECCO-Darwin ». Mikaloff Fletcher et al (2007) showed a natural carbon flux toward the Pacific whereas Mikaloff Fletcher et al (2006) presented a Cant flux toward the Indian Ocean. Could you comment ?

Thank you for pointing out those studies, we have now added some text to the discussion comparing what we find with their results: "A possible impact of the lack of mass transport constraint was seen in the large counter-clockwise circulation of carbon on Fig. 8 between the Southern Ocean, South Pacific, and Indian Ocean. These transports do not seem consistent with, for example, a westward Indonesian Throughflow (ITF) $C_{anth}$ transport of 0.05 Pg C yr$^{-1}$ as reported by Mikaloff Fletcher et al. (2006) or an eastward ITF $C_{nat}$ transport of 0.1 Pg C yr$^{-1}$ as reported by Mikaloff Fletcher et al. (2007). We have diagnosed a transport of $C_*$, which contains both $C_{anth}$ and a portion of $C_{nat}$, but the OTM transports nonetheless do not appear reconcilable with the ITF estimates".

C-07: Line 350: « the method therefore does not resolve either tracer or flux gradients within a water mass. The effect of the latter is illustrated on Fig. 5, where we compare the unaltered ECCO-Darwin boundary carbon fluxes with the result of binning the fluxes into water mass space and then remapping them back into geographical coordinates using a mask ». Maybe specify where the large differences occur: e.g. NE-PAC, SE-PAC, SO-ATL and Indian (source versus sink ?). Would those differences be the same when applying OTM with observations?

We have added some explanation: "There are differences, for example off the west coast of North America where ECCO-Darwin has outgassing (Fig. 5a) and the remapped fluxes show uptake (Fig. 5b); a similar situation off the west coast of the southern tip of South America; and also in the north of the Indian Ocean. Fig. 5b is the closest we can get to the true model fluxes with this configuration (with the caveat noted in section 2.4 that we have remapped surface fluxes into three-dimensional water masses)".

We also now note that "a closer match to the 'true' field may be achieved by combining the remapped term q(x, y, z)$^{adjust}$ with the true fluxes; therefore here we present a worst-case scenario where the detail of the true fluxes is not assumed to be reliable". This alternative method is a recent development that was not applied during the analysis for this paper.

C-08: Line 370: « for example in the case of the South Pacific/Indian Ocean, it could be beneficial to further split the Southern Ocean in a manner that allows the imposition of an Antarctic Circumpolar Current. ». A suggestion for future analysis: select the Drake Passage for the transport using observations available at this boundary (Meredith et al, 2011; Munro et al, 2015)?

Thank you for the suggestion, indeed such constraints may be useful!

C-09: Line 407: « We are working on producing our own global, full-depth, time-evolving estimates of DIC and $C_*$ in the ocean, using machine learning with satellite and GLODAP data, which we hope by combining with OTM will enable us to produce the first global estimate of the uptake, transport and storage of carbon directly from observations. ». Why not starting/testing OTM using MOBO-DIC (Keppler et al)? Is the MOBO-DIC period 2004-2017 too short to test OTM and because MOBO-DIC is

not extended to the bottom?

We added some further explanation about the potential of application using MOBO-DIC earlier in the paragraph, as follows: "....Unfortunately, these two estimates are limited, respectively, to the top 1500 m of the ocean, and to the Southern Ocean only. MOBO-DIC could form the basis for an application of OTM, but it would be necessary to somehow extend it to full depth."

Is your new global data-based product already developed? Could you specify the data that would be needed for applying OTM (T, AT, DIC, O2, nutrients, other ?).  Are the existing data synthesis available enough for your future analysis or would you recommend to extend GLODAP, SOCAT, etc... ?

We have made significant progress with the application of OTM to observations in the last few months, including in the machine learning reconstructions which have been validated and on which a paper is in preparation. We modified part of the final paragraph of section 4.2 as follows: "We are developing our own global, full-depth, time-evolving reconstructions from 1990-present of DIC and $C_*$ in the ocean that we hope to combine with OTM in future work. The reconstructions use deep neural networks trained on GLODAP DIC, Total Alkalinity, and nutrient data, with predictors of temperature and salinity from EN4, location, depth, and atmospheric $CO_2$ concentration."

C-09: Figure 2: curiosity: what, where are the outliers at high salinity 38 (Red Sea, MedSea, Arabian Sea?)

This is the Mediterranean (see e.g. https://journals.ametsoc.org/view/journals/phoc/42/5/jpo-d-11-0139.1.xml Fig 5).

C-10: Figure 5: In the legend maybe recall that (b) is for Case 2.

We have added this detail; the caption now reads: "Boundary carbon fluxes (air-sea CO2 flux - sediment flux) for the ECCO-Darwin 1995-2015 time-mean (a) and the BSP-binned fluxes remapped back into geographical coordinates (b). Note that (b) is the 'ECCO Darwin' flux against which the prior and solution are compared on Figure 4 and is also the prior for case 2."

C-11: Title: "An optimal transformation method applied to diagnosing the ocean carbon sink."

As there are also sources in some regions (EqPAC), maybe change the title: "An optimal transformation method applied to diagnosing the ocean carbon budget".

Thank you for the title suggestion; we agree it is an improvement and have adopted it.